# Pheochromocytomas and Abdominal Paragangliomas: A Practical Guidance

**DOI:** 10.3390/cancers14040917

**Published:** 2022-02-12

**Authors:** Jan Calissendorff, Carl Christofer Juhlin, Irina Bancos, Henrik Falhammar

**Affiliations:** 1Department of Endocrinology, Karolinska University Hospital, 171 76 Stockholm, Sweden; henrik.falhammar@ki.se; 2Department of Molecular Medicine and Surgery, Karolinska Institutet, 171 76 Stockholm, Sweden; 3Department of Oncology-Pathology, Karolinska Institutet, 171 65 Stockholm, Sweden; christofer.juhlin@ki.se; 4Department of Pathology and Cancer Diagnostics, Karolinska University Hospital, 171 76 Stockholm, Sweden; 5Division of Endocrinology, Diabetes, Metabolism, and Nutrition, Department of Internal Medicine, Mayo Clinic, Rochester, 55905 MN, USA; bancos.irina@mayo.edu

**Keywords:** pheochromocytoma, paraganglioma, genetics, imaging, histopathology, prognosis

## Abstract

**Simple Summary:**

Pheochromocytomas and abdominal paragangliomas (PPGLs) are rare. They can be discovered incidentally by imaging with computed tomography or magnetic resonance imaging and during hormonal surveillance in patients with known genetic variants that are associated with PPGLs. As most PPGLs are functioning, a hormonal work-up evaluating for catecholamine excess is recommended. Classical symptoms, such as tachycardia, hypertension and headache, can be present, but when the PPGL is discovered as an incidentaloma, symptoms may be lacking or be more discrete. PPGLs carry malignant potential, and patients should undergo close surveillance, as recurrence of disease or metastasis may develop. Genetic susceptibility for multifocal disease has gained more attention, and germline variants are commonly detected, thus facilitating detection of hereditary cases and afflicted family members. Any patient with a PPGL should be managed by an expert multidisciplinary team consisting of endocrinologists, radiologists, surgeons, pathologists and clinical geneticists.

**Abstract:**

Pheochromocytomas and abdominal paragangliomas (PPGLs) are rare tumors arising from the adrenal medulla or the sympathetic nervous system. This review presents a practical guidance for clinicians dealing with PPGLs. The incidence of PPGLs has risen. Most cases are detected via imaging and less present with symptoms of catecholamine excess. Most PPGLs secrete catecholamines, with diffuse symptoms. Diagnosis is made by imaging and tests of catecholamines. Localized disease can be cured by surgery. PPGLs are the most heritable of all human tumors, and germline variants are found in approximately 30–50% of cases. Such variants can give information regarding the risk of developing recurrence or metastases as well as the risk of developing other tumors and may identify relatives at risk for disease. All PPGLs harbor malignant potential, and current histological and immunohistochemical algorithms can aid in the identification of indolent vs. aggressive tumors. While most patients with metastatic PPGL have slowly progressive disease, a proportion of patients present with an aggressive course, highlighting the need for more effective therapies in these cases. We conclude that PPGLs are rare but increasing in incidence and management should be guided by a multidisciplinary team.

## 1. Introduction

Neuroendocrine tumors derived from the adrenal medulla and paraganglia are termed pheochromocytomas and paragangliomas (PPGLs). Pheochromocytomas are adrenal tumors arising from chromaffin cells of the adrenal medulla. PPGLs most often secrete catecholamines [1], and the paragangliomas are further subdivided into tumors from the sympathetic nervous system (functional tumors most often localized inferior of the diaphragm) and parasympathetic nervous system represented by head and neck paragangliomas, which are rarely functional. Catecholamines are secreted from PPGLs continuously or intermittently, potentially giving rise to typical symptoms as tachycardia, pallor, hypertension and headache. Such symptoms can be present, but in the current age PPGLs most often present as adrenal incidentalomas in about two-thirds of subjects [2,3]. Paragangliomas represent 15–20% of catecholamine-producing tumors [4].

PPGLs may be sporadic or found in a context of hereditary syndromes. All PPGLs exhibit malignant potential [5,6], with low frequency of metastatic cases (<5%) in patients with hereditary forms such as von Hippel-Lindau disease (*VHL*) and multiple endocrine neoplasia type 2 (MEN2 with *RET* gene variants), and increased risk of disseminated disease in patients with *SDHB* mutations [7]. Germline variants in a defined group of PPGL susceptibility genes are found in 40% of patients with PPGLs, making the tumor form the most heritable of all human neoplasias. Somatic mutations in a PPGL susceptibility gene are also common in PPGL [8,9], and found in up to 30% of cases [10,11].

As of today, a large number of PPGL susceptibility genes have been discovered [12], in which a mutational event on the germline level may confer an increased risk of developing PPGLs. Thus, for these patients, genetic counseling is important to guide follow-up and to find relatives with potential germline variants [13].

Head-neck paragangliomas (HNPGLs) are parasympathetic and most often non-functional [13]. HNPGLs are most often located in the glomus caroticum (60%) but can also be discovered in paraganglia along the vagal nerve, the bulbus jugularis and in the tympanic region [8]. These lesions will not be further discussed in this review. In this review, PPGLs denote pheochromocytomas and abdominal paragangliomas, i.e., lesions inferior to the diaphragm (Figure 1).

The aim of this review is to present current knowledge of PPGLs, including epidemiology, clinical work-up, histopathological findings, and treatment, including special situations, metastatic disease, and genetics affecting the risk for metastatic disease, with implications for follow-up.

## 2. Materials and Methods

Literature searches in PubMed were performed with the terms “pheochromocytomas” and “paragangliomas” in combination with other terms such as “prognosis”, “cerebrovascular disease”, “metastatic disease”, “genetics” and “imaging”. Articles relevant to these searches was also identified in the reference lists of the retrieved articles and in the authors´ personal files.

## 3. Results

### 3.1. Epidemiology

In a population setting, standardized incidence rates of PPGLs were reported to have increased 3–5-fold over the last 40 years, from 1.4–2.1 per million person-years in 1970s–1980s to 5.7–6.6 per million person-years in the period 2010–2015 [14,15]. The increase in PPGL incidence was explained mainly by an increase in incidentally discovered PPGLs of smaller size (<4 cm) in older patients (>50 years) [14,15]. The increase in PPGL incidence could be explained by a true increase in the development of PPGLs, better detection of PPGLs due to an increase in imaging performed, as well as improved and more often used biochemical tests, or possibly a combination of these factors. For example, in a population study conducted in Olmsted County (MN, USA), the incidence of adrenal tumors increased 10-fold between 1995 and 2017, in parallel with and proportional to the increase in the number of abdominal computed tomography (CT) and magnetic resonance imaging (MRI) scans performed during the same period [16]. A true increase in PPGL incidence is probably less likely. However, if we speculate, changes in tumorigenesis could be responsible for such an increased incidence, and environmental factors may play a role.

The number of patients with evaluated PPGLs was also reported to have increased. For example, at Mayo Clinic (Rochester, MN, USA), the mean number of patients with new diagnoses of pheochromocytoma evaluated between 2005–2016 was 24.7/year versus 8.6/year between 1971 and 1980 [3]. In the same study, the proportion of patients diagnosed with PPGLs incidentally was reported to have increased from <10% in the 1970s to 31% in 1995–2004, and 60% in 2005–2016 [3]. This is similar to the population-based study conducted in Denmark (12.2% in 1970s vs. 59% after 2007) [14]. As genetic testing has become more accessible and popular, the proportion of patients diagnosed as a part of genetic screening has also increased, representing around 5–12% of patients discovered with PPGLs in the last decade [3,14,17]. A higher proportion of patients discovered with PPGL incidentally or based on genetic case detection testing [2,3,14,15] over the years explains the reported decrease in the median tumor size at which PPGLs are diagnosed. It could be speculated that in the future, even more PPGLs will be discovered as incidentalomas or via screening of relatives following a certain genetic finding in a proband.

Most studies report no sex or a slight female predominance of PPGLs, with 50.5–57% of patients with PPGLs being women [3,14,15,17,18]. The median age at the time of PPGL diagnosis was reported to be between 48 and 55 years, younger in those diagnosed based on genetic case detection testing (median age of 30–42 years) [3,15,17,18,19,20,21,22,23]. The majority of patients with PPGLs are diagnosed with unilateral pheochromocytoma (81–89%) or single paraganglioma (7–18%), while multifocal PPGLs (<1–4%) and metastatic PPGLs (3–15%) are rare [3,14,15,17].

### 3.2. Clinical Presentation

The presenting symptoms of PPGLs vary extensively, and these can be present in many other clinical conditions [24]. For this reason, PPGLs are often called “the great mimic”, and significant delays of diagnosis are not uncommon [14,25].

Previously, most PPGLs were discovered due to symptoms and signs suspected to be related to catecholamine excess, such as paroxysmal hypertension and the classic triad of headaches, sweating and palpitation.

The classical triad is in fact nowadays not that common, with only a fifth of patients having all three components [2,14], but most patients have a least one of the three symptoms. Around one- to two-thirds have paroxysmal symptoms [2,14]. In one study of 92 patients with PPGLs, the symptoms and signs described were palpitation (53%), anxiety (46%), sweating (41%), headaches (37%), tiredness (28%), orthostatic symptoms (27%), feeling hot/flush (24%), nausea (22%), weight loss (16%), pallor (12%) and no symptoms at all (9%) [2]. Those found due to suspicion of PPGL had more paroxysmal symptoms, palpitations, headaches, pallor, nausea, tremor and orthostatic symptoms compared to those found as an incidentaloma. Those with suspicion of PPGL also had more different symptoms than the incidentaloma group, while those screened due to a familial syndrome had the fewest symptoms [2]. Very rarely, patients can present with symptoms and signs of Cushing’s syndrome due to ACTH/CRH secretion from the PPGL [26]. Adrenal medullary hyperplasia gives a presentation similar to that of PPGL, though with fewer symptoms, and almost half of these are found due to catecholamine screening in patients with known genetic variants [27]. Adrenal medullary hyperplasia can also present as Cushing’s syndrome [28]. Another rare combination is pheochromocytoma in a patient with concomitant primary aldosteronism [29].

Cardiovascular manifestations are common in patients with PPGLs and are due to longstanding hypertension in those with undetected PPGLs or episodes of profuse catecholamine excess [30]. Severe hypertensive crisis can lead to cardiovascular complications. One-fifth to more than one-third of patients with PPGLs have cardiovascular complications [2,31].

Paroxysmal or chronic hypertension is present in up to 95% of patients with pheochromocytomas [2,32,33]. The blood pressure typically fluctuates between hypertension and hypotension in cyclic attacks [32,34]. PPGL-induced hypertensive and multisystem crisis is feared, resulting in cardiovascular and cerebrovascular complications [34,35,36]. Takotsubo syndrome, a suddenly stunned or weakened myocardium, can occasionally be the initial presentation of PPGL [37,38]. Cardiovascular symptoms, e.g., chest pain or dyspnea and sometimes abdominal pain together with symptoms and signs of PPGL, could indicate a PPGL-induced Takotsubo syndrome [39]. Dilated cardiomyopathy or catecholamine-induced cardiomyopathy has been reported to be an important complication of PPGLs, occurring in up to 11% of cases [40]. In the majority, cardiomyopathy was transient if the PPGL was diagnosed early and treated correctly [41] but could be progressive and fatal if the tumor remained undiagnosed [42,43]. Some of these cases with cardiomyopathy probably suffered from undiagnosed Takotsubo syndrome [30].

### 3.3. Biochemical Diagnosis

The Endocrine Society Clinical Practice Guideline recommends that biochemical evaluations be performed with plasma free metanephrines or urinary fractionated metanephrines [4]. Liquid chromatography with mass spectrometric (LC-MS/MS) or electro-chemical detection is recommended. Raised metanephrine levels are the hallmark of a functional PPGL, but non-functional tumors will be missed if relying solely on measured catecholamines. Both urine and plasma tests are accurate, but no head-to-head evaluations have been performed. In a recent meta-analysis comparing the plasma and urine metanephrines, supine plasma was proved to be more sensitive and displayed higher specificity than 24-h urine samples [44]. Plasma metanephrines measured after 20 min bed rest have improved accuracy. If LC-MS/MS is performed, patients should ideally not use caffeine or tricyclic antidepressants, as this may result in false elevated metanephrines, both in urine and plasma [4]. In methods using liquid chromatography with electrochemical or fluorometric detection (LC-ECD), paracetamol, mesalamine and sulfasalazine also can give false elevation [45]. The combination of elevated metanephrine and normetanephrine levels, in urine or plasma, is more often noted in adrenal tumors; especially significant metanephrine level elevation indicates a pheochromocytoma [46], whereas isolated increases in normetanephrine levels can be found in both pheochromocytomas and paragangliomas. Notably, the biochemical profile also depends on a particular hereditary syndrome associated with PPGLs. For example, patients with cluster 1 mutant tumors (*VHL/SDHx*) differ from those with cluster 2 kinase-driven PPGLs (*NF1/RET*) in expression of the phenylethanolamine N-methyltransferase (PNMT) enzyme responsible for the conversion of norepinephrine to epinephrine. Cluster 1 noradrenergic tumors, therefore, show little/absent PNMT, with strong increases in normetanephrine with no or relatively small increases in metanephrine, while cluster 2 adrenergic tumors show tumor-derived increases in plasma metanephrine [47]. A value > 2 times the upper normal level is suggestive for a functional tumor [48]. However, only slightly elevated levels or even levels within normal ranges (“biochemically silent” pheochromocytomas) can be found more often in hereditary cases undergoing screening and in smaller pheochromocytomas [3,23]. Thus, in a tumor detected by CT/MRI, any level of catecholamines or metanephrines/methoxytyramine above the reference limit could be considered indicative of a functional tumor. Pheochromocytomas and abdominal paragangliomas are rarely non-functional, and the diagnosis is first made on imaging, or rarely by biopsy or post-operatively by histopathological examination [49,50]. Even though these tumors do not produce catecholamines/metanephrines, they may secrete dopamine or its metabolite methoxytyramine, which can be measured [51]. Chromogranin A (CgA) and plasma methoxytyramine can be added, the latter in case of suspicion of malignancy. CgA is found in most neuroendocrine tumors, and also in local PPGLs, but in the latter this test is inferior to metanephrines, although recommended by some [52]. Rather, falsely elevated levels may be observed in patients treated with proton pump inhibitors, or in conditions such as atrophic gastritis and inflammatory bowel disease [53]. Thus, positive results in such settings should be critically interpreted. As metastatic PPGLs may lack the enzymatic capacity to secrete catecholamines, increased levels of methoxytyramine can be measured in such instances [51,54]. Elevation of plasma methoxytyramine levels, solitary or together with other plasma metanephrines, has been found in up to 70% of patients with pheochromocytomas arising in the setting of *SDHB* and *SDHD* variants [55].

### 3.4. Imaging

In clinical practice, pheochromocytomas are usually detected by CT (Figure 2), and following hormonal work-up reveals catecholamine excess [4]. Pheochromocytomas are often well-defined tumors with unenhanced attenuation measurements of 30–40 Hounsfield units [21,56,57,58]. On CT scans, pheochromocytomas may on occasion be cystic and display features with calcification, fibrosis and internal hemorrhage [21]. As pheochromocytomas were reported to present as a composite tumor (along with an adrenal adenoma or ganglioneuroma) [21,59,60], imaging characteristics may reflect two distinct areas with different attenuation or heterogeneity. In targeted adrenal investigations, the differential diagnosis of lipid-rich adrenal cortical adenoma can most often be ruled out, as pheochromocytomas exhibit Hounsfield units > 10. However, pheochromocytomas cannot be distinguished accurately from adrenal metastases or adrenal cortical carcinomas with unenhanced CT, as all of these masses are lipid-poor, with most demonstrating Hounsfield unit measurements >20. If only normetanephrine levels are elevated, an extra-adrenal lesion can be suspected, and whole-body imaging should be performed [61].

MRI can delineate most adrenal masses [62]. Pheochromocytomas are classically light bright lesions on T2 weighted imaging, but a high intensity or intermediate signal is more common. With MRI, vascular invasion could be assessed prior to surgery and could be suspected in large tumors [63]. CT and MRI are often sufficient as localizing tools when biochemistry clearly indicates the presence of PPGL.

When imaging features are unclear or negative with CT/MRI, including during surveillance, various methods of nuclear imaging can be used. Nuclear imaging is more sensitive than imaging with CT or MRI [64]. ^123^I-metaiodbenzylguanidine (MIBG) can confirm a pheochromocytoma [65]. Other functional imaging methods recommended by the European Association of Nuclear Medicine (EANM) are with PET ligands such as ^18^F-fluorodopamine, ^18^F-dihydroxy-phenalalinine (DOPA), ^18^F-FDG or ^68^Ga-DOTATOC/DOTATATE/DOTANOC [61]. The tracer that is most often available is ^18^F-FDG [66]. The sensitivity with ^18^F-FDG is 80–100% for a pheochromocytoma, but the specificity is low and even less in extra-adrenal disease. ^18^F-FDG is also less precise in RET-related pheochromocytomas. It has high accuracy for *SDHB-*positive, but less for *SDHB-*negative, tumors with 83% and 62% sensitivity, respectively [66]. Ga-DOTATOC and ^18^F-DOPA have been shown to be more accurate than ^123^I-MIBG [67], especially in small lesions and in extra-adrenal disease [68,69].

It has been suggested that nuclear modalities should be performed in all lesions >5 cm in size and in all extra-adrenal PPGLs [70]. Current expert opinions on nuclear imaging provide guidance on which PET to be applied depending on the genetic status in these patients, if known [61]. In PPGLs with *NF1/RET/VHL/MAX* variants, ^18^F-DOPA has been suggested to be the first choice, followed by ^123^I-MIBG, whereas in *SDHx*-related disease and in extra-adrenal cases, ^68^Ga-DOTATOC should be the first choice, followed by ^18^F-FDG [61].

In metastatic disease, nuclear imaging is more often successful in localizing disease [61], which is discussed later in Section 3.9.

### 3.5. Histopathology

Pre-operative fine-needle aspiration biopsy (FNAB) or core-needle histological evaluations of pheochromocytomas are, in general, restricted to pinpoint metastatic deposits, and these methods have little value in the primary work-up of an adrenal incidentaloma [71]. As potentially life-threatening complications may follow biopsies of highly vascular, catecholamine-secreting PPGLs, prior biochemical testing is strongly recommended [72]. The histological assessment of PPGLs is based on morphology and immunohistochemistry, with little or no use of molecular testing to aid in the diagnostic process. Usually, PPGLs are nest-forming tumors with a highly vascularized stroma (Figure 3).

Tumor cells exhibit a granular and slightly basophilic cytoplasm, and nuclei can vary in size and appearance, with either mono- or pleomorphic features [13,73]. Immunoreactivity to classic neuroendocrine markers such as CgA and synaptophysin is almost always present, and second-generation neuroendocrine markers ISL LIM Homeobox 1 (ISL1) and INSM1 have also been found as reliable markers of an adrenal medullary origin in pheochromocytoma [74,75,76] (Figure 3). Additionally, the transcription factor GATA3 is useful in identifying PPGLs and excluding other neuroendocrine neoplasms [77]. A S100 stain can help to visualize the sustentacular cell network that is often present in PPGLs and normal adrenal medulla [78,79].

The current World Health Organization (WHO) classification of endocrine tumors from 2017 launched the concept that all PPGLs carry malignant potential, and advocated a change of nomenclature; from “benign vs. malignant PPGL” to “non-metastatic vs. metastatic PPGL”, with metastatic PPGL defined as the finding of chromaffin tumor cells in a non-chromaffin site (regional lymph nodes or distant sites) [13]. However, although this diagnostic criterion obviously displays an exceedingly high sensitivity towards true, malignant disease, the need for an algorithm predicting future metastatic spread in PPGLs limited to the primary site is still considerable. There have been a few attempts to pinpoint metastatic potential through histology and/or immunohistochemistry, most notably the Pheochromocytoma of the Adrenal gland Scaled Score (PASS) [80]. PASS is a strictly histology-based algorithm for pheochromocytoma, incorporating 12 parameters in which a score of 4 points or more indicates an increased risk for malignancy. Although corroborated by some investigations [81,82], others have not found PASS useful to identify metastatic potential [83]. The PASS score also exhibits high inter-observer variability [83,84]. Another algorithm, the Grading system for Adrenal Pheochromocytoma and Paraganglioma (GAPP), incorporates both pheochromocytomas and paragangliomas and combines histological features, the Ki-67 proliferation labeling index and biochemical findings [85]. Although recent, the value of this algorithm has been validated by others [84]. Moreover, in a recent meta-analysis of both scoring systems, both the PASS and GAPP scores were found to be excellent in ruling out metastatic potential rather than ruling in the same parameter [86]. Thus, these histological triaging systems might have value in pinpointing PPGLs with exceedingly low risk of future dissemination.

### 3.6. Genetics and Molecular Immunohistochemistry

As detailed above, histopathological examination may not be sufficient to determine the true metastatic potential of a PPGL. However, next-generation sequencing is slowly paving its way into clinical routine practice, making it possible to interrogate the somatic mutational status of each PPGL. Moreover, complementary immunohistochemical analyses have been proposed to predict the underlying genetics of each lesion (termed “molecular immunohistochemistry”). The molecular background of PPGLs may appear intimidating to a clinician not used to genetics, as PPGLs constitute the most hereditable of all human tumors, with up to 30 susceptibility gene variants on the germline level and an additional set of genetic aberrancies occurring on the somatic level. The precise mechanisms regarding each gene variant and the development of PPGLs in both sporadic and syndromic settings have been reviewed elsewhere [73,87], and this review focuses on the clinical utility of knowing which gene aberrancy is at play in different PPGLs, and how this could affect the prognostication.

In short, PPGLs adhere to one out of four main molecular clusters based upon their general expressional profiles when the messenger RNA (mRNA) output is assessed: cluster 1 (pseudo-hypoxia driven PPGLs), cluster 2 (kinase driven cluster), cluster 3 (Wnt driven cluster) and cluster 4 (cortical admixture cluster). Cluster 1 can also be further divided into two different sub-groups: one tricarboxylic acid (TCA) cycle aberrant and one TCA cycle non-aberrant group. As a general principle, genetic variants in certain PPGL susceptibility genes (irrespective of whether they are somatic or germline) will cause the tumor to express an mRNA signature adhering to one of these four clusters [88].

Cluster 1 PPGLs include pheochromocytomas and abdominal paragangliomas carrying the highest risk of metastatic spread, especially the TCA cycle aberrant tumors carrying genetic variants in enzymes regulating the TCA cycle, including *SDHA, SDHB, SDHAF2, FH, MDH2, ISH1* and *SLC25A11* [73]. A faulty TCA cycle will lead to an accumulation of onco-metabolites such as succinate, fumarate and alpha-ketoglutarate derivatives, which in turn may activate hypoxia inducible factor alpha (HIFs) and may interfere with the epigenetic machinery. This, in turn, may drive PPGL formation [89]. Variants in other genes involved in hypoxia may also drive the development of PPGLs, but in a TCA cycle non-aberrant fashion. These genes include *VHL, EPAS1/HIF2-alpha, PDH1/EGLN2* and *PDH2/EGLN1*. Somatic and germline variants in these genes also activate HIFs, but do not affect the epigenetic machinery to the same extent as TCA cycle gene variants do. Therefore, the metastatic potential of the TCA cycle non-aberrant cluster 1 PPGLs is in general lower than that for TCA cycle aberrant cases in the same cluster, with the exception of *EPAS1* [90]. Cluster 2 PPGLs are built-up mostly by pheochromocytomas driven by variants in genes regulating kinase-driven pathways, including *NF1, KIF1B, MAX, RET, TMEM127* and *H-RAS* [91]. The metastatic potential of PPGLs adhering to this cluster is generally low, with the exception of MAX, in which an intermediate risk of dissemination is noted. Cluster 3 PPGLs contain pheochromocytomas that are overrepresented in somatic *UBTF-MAML3* fusion genes and *CSDE1* mutations [88]. This cluster is also coupled to metastatic potential, and *MAML3* has been shown to drive tumor progression and increase invasive behavior in vitro [92]. Finally, cluster 4 contains pheochromocytomas with an exceedingly low risk of metastatic disease. These tumors are mostly driven by somatic *NF1* or germline *RET* variants, and the biological meaning of the observed mRNA admixture with adrenal cortical elements is not known [93].

The three main genetic groups of PPGLs with associated markers of clinical interest are illustrated in Figure 4.

While all PPGL patients should undergo genetic screening in order to pinpoint an eventual germline event, which may lead to the discovery of a hitherto unknown syndrome in a specific proband, there are also ancillary tests performed on the tumor tissue that may help to triage these tumors further. For example, SDHB immunohistochemistry is an efficient tool to screen for underlying *SDHx* gene family mutations, as the immunoreactivity is lost if the SDH complex is disrupted [94]. This is useful, as the stain may help identify TCA cycle aberrant cases with an increased risk of metastatic spread. On a similar note, CAIX staining may identify cluster 1 TCA cycle non-aberrant cases with *VHL* gene variants, as CAIX is up-regulated in the presence of a *VHL* gene aberrancy [95,96]. Finally, alpha-inhibin may serve as a marker for cluster 2 PPGLs, but this finding needs to be reproduced before being taken into clinical consideration [96]. Moreover, genetic screening of somatic DNA including mutations in *SDHB*, *ATRX* and *SET2D*, *MAML3* fusions as well as *TERT* gene aberrations may also help identify potential poor-prognosis cases prior to dissemination [91,97,98] (Figure 5).

### 3.7. Management

Multidisciplinary teams involving surgeons, endocrinologists, radiologists including nuclear medicine physicians, clinical geneticists and pathologists, all well acquainted with PPGLs should be involved in the care of the patient to assure personalized management and improved outcomes [4]. A scheme of the diagnostic procedure and follow-up is presented in Figure 6.

Surgical resection is the mainstay of therapy for any patient with a localized PPGL [6]. Surgery could be performed with minimally invasive (e.g., laparoscopic) adrenalectomy in adrenal lesions <6–7cm, while open surgery is usually reserved for larger tumors [4]. Surgery for paragangliomas can be more difficult, so open resection may be needed depending on surgical expertise and tumor location. However, laparoscopic resection is often possible for small, noninvasive paragangliomas in surgically accessible locations [99]. In a patient with bilateral pheochromocytoma, partial adrenalectomy can successfully preserve adrenal function in more than 75% of patients but does present some risk of local recurrence [20,22].

We recommend pre-operative medical therapy with alpha- and secondary beta-blockade in any patient scheduled to have surgery for suspected pheochromocytoma or paraganglioma (Table 1). Medical therapy with alpha-blockade is initiated first, usually 10–14 days prior to planned procedure, though some centers titrate more slowly (1–3 months) prior to the surgery to reduce side effects [17]. Some patients are well-controlled with only low dose alpha-blockade. The goal of therapy is to reach a low normal systolic blood pressure and a heart frequency <80–90 bpm. A 24-h blood-pressure evaluation could be performed 3–4 days prior to procedure in order to be able to optimize alpha-blockade. Presence of mild orthostasis is usually an indicator of optimal alpha-blockade. Patients are usually educated on the orthostatic vital measurement and are advised to call in daily, and to have measures of blood pressure and pulse at home, which aids in titration via telephone with their physician, to achieve optimal titration of alpha-blockade.

Patients need also to be advised on the potential side effects of alpha-blockage, such as postural hypotension, tachycardia, nasal congestion, fatigue and somnolence. Optimal fluid and increased salt intake usually counteract some of these side effects. The initial dosing, incremental increase in the medication, and the final dose depend on individual characteristics (such as the degree of hypertension, the tumor size and the degree of catecholamine excess, as well as development of any side effects).

If titration with phenoxybenzamine/doxazosin results in inadequate blood pressure control, beta-blockers and/or calcium-channel blockers should be added. Beta-blockade is usually initiated at least 5–7 days after alpha-blockade. It is important to note that in patients with PPGLs, beta-blockers should not be initiated prior to alpha-blockers due to challenges in cardiovascular management as a result of unopposed alpha blockade [32]. Side effects to beta-blockage include bradycardia and hypotension, but are unlikely in the setting of daily titration and short duration of therapy, or if doses are increased with intervals of a week. Calcium channel blockade is usually used in special circumstances, such as intolerance to alpha-blockade, or the need for an additional agent to reach the target blood pressure.

Metyrosine, a tyrosine hydroxylase inhibitor that blocks the rate-limiting step in catecholamine synthesis, is usually used in patients at high risk for catecholamine release during the procedure despite alpha blockade, or those intolerant to alpha-blockers [100]. Special situations when metyrosine could be used include anticipated difficult resection or patients undergoing thermal ablation with anticipated significant catecholamine release from the treated metastasis [100,101]. In these cases, the protocol used at Mayo in 122 cases with PPGL is described in Table 1. Other studies reported using doses from 600 to 4000 mg per day, with mean doses of 1000–2000 mg for less-complicated situations [100]. However, this is an expensive agent, as 30 capsules cost more than USD 11,000, and it is not available in all countries.

Either phenoxybenzamine or doxazosin can be used, depending on availability and cost. When comparing pretreatment with phenoxybenzamine and doxazosin in a randomized controlled trial, no difference in the duration of blood pressure outside the target range during resection of PPGL-tumor was found [102]. Although phenoxybenzamine prevented intraoperative systolic blood pressure above the target range and hemodynamic instability better than doxazosin, there was no difference in side effects, complications or length of hospital stay between the two groups. In a hypertensive crisis, an intravenous alpha-blocker such as phentolamine is preferred over oral alpha-blockers due to its rapid action [17,103].

### 3.8. Follow-Up

Several weeks after surgery, a follow-up with biochemical tests should be performed to show successful removal of the tumor. Thereafter, lifelong annual follow-up with biochemical tests is generally recommended to discover recurrent or metastatic disease [4]. However, follow-up should be personalized, taking into consideration especially the genetic status.

Patients with sporadic PPGLs without any underlying genetic variant should be offered surveillance with clinical investigation and metanephrines at least for 10 years, and life-long if there are additional risk factors [52,104].

Cluster 1 PPGLs require a closer follow-up due to the risk of metastasis and recurrence as well as other manifestations of the familial syndrome [48]. For those with a PPGL and *SDHA/B, HIF2A/EPAS*- and *FH* mutations, who have the greatest risk of metastasis, biochemical tests every 6 to 12 months and imaging every 1 to 2 years have been advised [48,104]. Since patients with PPGLs and *SDHC/D/AF2*- or *VHL* mutations have a lower risk of metastasis, it is probably enough to perform biochemical tests every 12 months and imaging every 2 to 3 years [48,105]. To minimize radiation exposure, MRI is recommended and should cover from the base of the skull to pelvis. However, since MRI is not good at detecting lung metastasis, alternating with low-dose chest CT in addition to MRI covering the base of the skull, neck, abdomen and pelvis is an option. Those with *VHL* mutations also have an increased risk of renal carcinomas and hemangioblastomas [106]; thus, an abdominal MRI every 12 months and a central nervous system MRI every 24–36 months should be considered [48].

In cluster 2 PPGLs, patients with high- or moderate-risk *RET* gene variants as well as *NF1, TMEM127* and *MAX* pathogenic variants could, in addition to yearly biochemical tests, have an abdominal/pelvic MRI every 3 years [48,104,107,108]. The optimal follow-up for cluster 3 PPGLs is unclear; however, since they have high recurrence and metastasis potential [90], follow-up could be similar to that for cluster 1 PPGLs [48].

A guideline from the European Society of Endocrinology recommended that all operated patients should be followed with biochemical tests for at least 10 years, and this applies to radically treated patients with no known pathogenic gene variant. Patients with extra-adrenal disease, large tumors and germline variants should be followed life-long [52].

Follow-up of asymptomatic carriers with genetic variants is recommended for *SDHx* carriers with metanephrines and also imaging with PET/CT. If this is negative, yearly control of blood pressure is recommended, with control of metanephrines every 2 years and imaging with whole body MRI every 2 to 3 years [48]. 

In children with *SDHB* probands, genetic testing should be performed at the age of 6–10, and at 10–14 years for carriers with other *SD* gene variants, with imaging intervals as for adults [48].

### 3.9. Metastatic Disease

All PPGLs are believed to harbor malignant potential, and approximately 10% to 15% of pheochromocytomas and up to 50% of abdominal paragangliomas will display metastatic disease [6,109]. Differential diagnoses constitute other malignancies, such as lymphomas and metastases from lung, kidney, GI tract, and breast cancer and malignant melanomas. In a recent European multicenter study of 169 patients with metastatic disease, 53% had a prior surgery for pheochromocytoma, elevated urine or plasma hormones were positive in 81%, a genetic *SDHB* variant was found in 42%, and the time from initial diagnosis to malignant disease was 43 months (0–614) [110]. New symptoms, imaging and/or biochemical tests will support suspicion of metastasis. EANM recommends ^68^Ga-DOTATOC/DOTATATE PET to be used as first choice in metastatic PPGL, while ^18^FDOPA can be the second choice in patients with no known *SDHB* mutation, or ^18^F-FDG PET the second choice in those with *SDHB* mutation, and ^123^I-MIBG scintigraphy is only used if ^131^I-MIBG therapy is considered [61]. Urinary dopamine, or even better, plasma methoxytyramine, which is a catecholamine metabolite, can indicate metastasis in patients with PPGL [54]. The likelihood of metastasis increases if the tumor is >5 cm and increases further to 50% in pheochromocytomas of more than 10 cm, and paragangliomas of more than 9 cm in size have an 80% likelihood of metastases [54]. The presence of *SDHB* mutations and younger age at initial presentation increase the malignant potential [111,112].

In patients with metastatic PPGL, metastases may be synchronous in 35–40%, or asynchronous in 60–65% of patients [113,114], developing at a median of 5.5 years following the initial diagnosis of PPGL [113]. Overall, most patients with metastatic PPGL present with a relatively indolent disease, with a median disease-specific survival of 33.7 years [113]. Several factors are associated with a rapidly progressive metastatic PPGL, including male sex, older age at the time of initial diagnosis, presence of synchronous metastases, larger tumor size, elevated dopamine/methoxytyramine levels, and not undergoing primary tumor resection [113,114]. Interestingly, the largest study to date has not demonstrated differences in progression of metastatic PPGL based on *SDHB* status [113]. In metastatic pheochromocytomas, metastases are usually found in bones and lymph nodes, whereas in paragangliomas, hepatic metastases are more frequent [113]. As most patients with metastatic PPGLs have indolent disease, local therapies that include surgical excision, thermal ablation, or external beam radiation can successfully treat lesions that are growing with little toxicity and few side effects [115,116,117]. Patients with metastases of PPGLs only to regional lymph nodes may be treated with surgery. Moreover, there may be a survival benefit for patients with distant metastases to have surgical removal of the primary tumor [113,118], but this is disputed [119]. If local recurrence occurs, repeated surgery can be performed. Intraoperative rupture of the tumor capsule can lead to recurrent and metastatic disease after many years [120]. Bone metastases occur in about 70% to 80% of patients with metastatic PPGL, sometimes resulting in severe pain, spinal cord compression, and pathologic fractures [121]. Treatment with bisphosphonates such as zoledronic acid or denosumab will improve quality of life, delay pain and skeletal complications and is recommended in all PPGL patients with metastatic bone disease [6,122]. Since treatment in metastatic PPGLs is essentially palliative and many metastatic PPGLs are slow-growing with an indolent course, a “wait and see” strategy under close surveillance is a good option [6,123]. However, if systemic treatment is needed due to progression, metanephrine secretion and symptoms such as uncontrolled hypertension or arrhythmia, there are some different options. Radionuclide treatment is the first-line treatment in patients with slowly progressing metastatic PPGLs, with ^131^I-MIBG [124,125,126] or ^177^Lu-DOTATATE [127,128,129] often being used, and these are the recommended choices depending on which shows best expression [6]. If both demonstrate equal uptake consideration for patient age, bone marrow and kidney function has to be made. ^131^I-MIBG could be used in patients with a good bone marrow reserve. In patients with elevated cardiac risk, ^131^I-MIBG might be preferred over DOTATOC, as this is not considered to be associated with increased secretion of metanephrines [130].

DOTATATE therapy might be slightly more effective than ^131^I-MIBG [131], and potentially more so in cluster 1 TCA cycle aberrant tumors [132]. In patients with rapidly progressive disease, chemotherapy is recommended. A combination of cyclophosphamide, vincristine, and dacarbazine has been used in some studies [133,134]. Temozolomide has sometimes been tried in metastatic PPGLs, with partial response in patients with *SDHB* pathogenic variant [135]. Other therapies, such as sunitinib [136], everolimus [137], and immune checkpoint inhibitors [138], have all demonstrated only modest effect. Local ablative therapies, e.g., radiofrequency ablation, cryoablation, percutaneous thermal ablation or external beam radiation, might yield local control or decreased symptoms in some patients [115,117].

### 3.10. Pregnancy in Patients with Pheochromocytomas and Paragangliomas

PPGL in pregnancy is rare, and many times goes unrecognized until later, potentially causing severe fetal and maternal complications of catecholamine excess [139]. In a recent combined retrospective multi-center study and systematic review of 249 pregnancies in 232 women with PPGL, only 15% of pregnancies were reported in women with known PPGL prior to conceiving [140]. Around half of the cohort were women discovered with PPGL during the course of pregnancy, and a third were diagnosed with PPGL only after and within 1 year after delivery or miscarriage. Overall, fetal or maternal complications occurred in 14% of cases, most in those with unrecognized PPGLs during pregnancy. Notably, no complications of catecholamine excess occurred in women with known PPGLs prior to conceiving, despite the higher rate of metastatic disease in this subgroup, possibly due to more intense monitoring during pregnancy, lower degree of catecholamine excess, and optimal management during pregnancy. Unrecognized PPGL (odds ratio of 27), abdominal or pelvic location of PPGL (odds ratio of 11), and severe catecholamine excess of at least 10 times the upper normal limit (odds ratio of 4.7) were associated with adverse outcomes [140]. Optimal alpha-blockade during pregnancy was associated with fewer complications (odds ratio of 3.6 for absence of alpha-blockade), suggesting that initiation, dose titration and optimal medical management during pregnancy is key in pregnant women with PPGL. In contrast, antepartum PPGL surgery was not shown to improve outcomes, though this could also be due to potential selection bias and confounding factors.

As genetic association of PPGL is more likely when PPGL occurs at a younger age, it is not surprising that the majority of women included in the study had either a familial history of PPGL or tested positive for one of the pathogenic variants of PPGL, most commonly *SDHx* variants *RET, VHL* and *NF1*. Appropriate counseling and case detection in women of reproductive age who are known carriers can allow appropriate planning for pregnancy [139]. When PPGL is discovered in a young woman without a known genetic predisposition, appropriate genetic testing should be offered, along with counseling.

## 4. Conclusions

PPGLs require a careful diagnostic and management approach by multidisciplinary teams specialized in radiology, genetics, surgery, endocrinology and pathology. Imaging is often the first tool indicating PPGL. A hormonal work-up will in most instances display elevated plasma or urine metanephrines. Genetic testing is recommended in all patients with PPGLs. A cure can be achieved with surgery, but all tumors may harbor malignant potential. A long-term biochemical and/or imaging follow-up is needed in all patients.

## Figures and Tables

**Figure 1 cancers-14-00917-f001:**
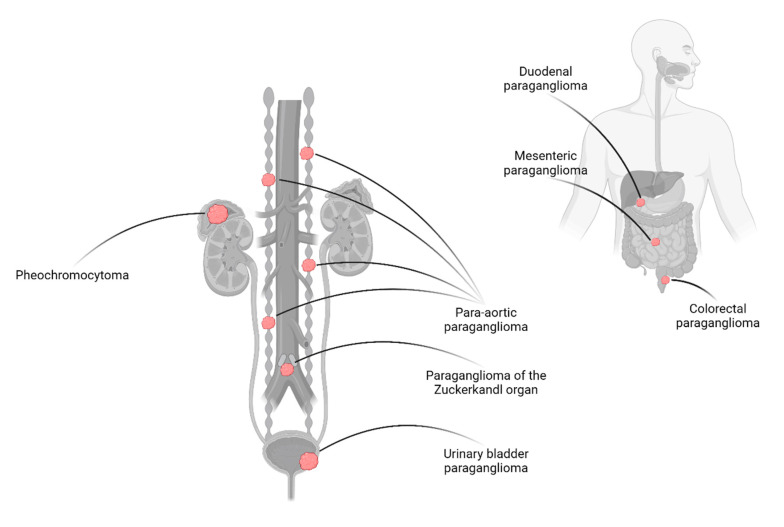
Anatomic overview of pheochromocytoma and the most commonly encountered abdominal paraganglioma. While pheochromocytomas arise in the adrenal medulla, abdominal pargangliomas derive from sympathetic paraganglia. The latter entity most commonly occurs in the retroperitoneum along the sympathetic trunk, not seldom within the organs of Zuckerkandl. Paragangliomas may also develop in paraganglia distributed along the urinary and gastrointestinal tracts. Created with BioRender.com.

**Figure 2 cancers-14-00917-f002:**
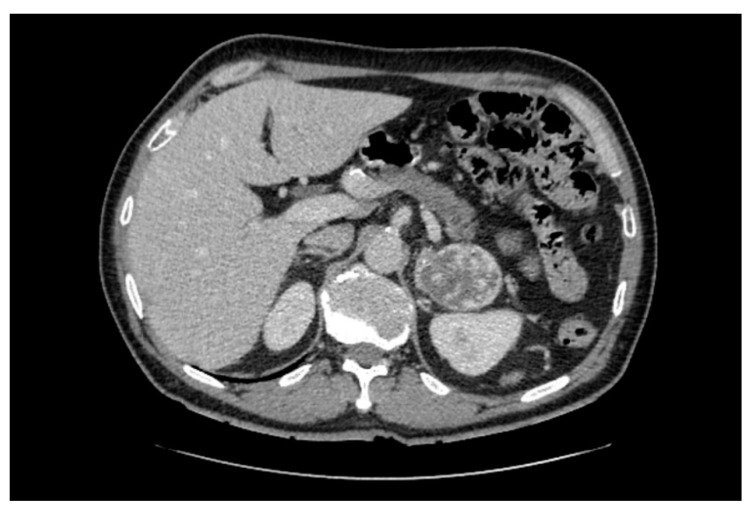
Computed tomography displaying a 6 × 5 cm large, well-defined and heterogenous-appearing pheochromocytoma originating from the left adrenal gland. The right adrenal gland is normal.

**Figure 3 cancers-14-00917-f003:**
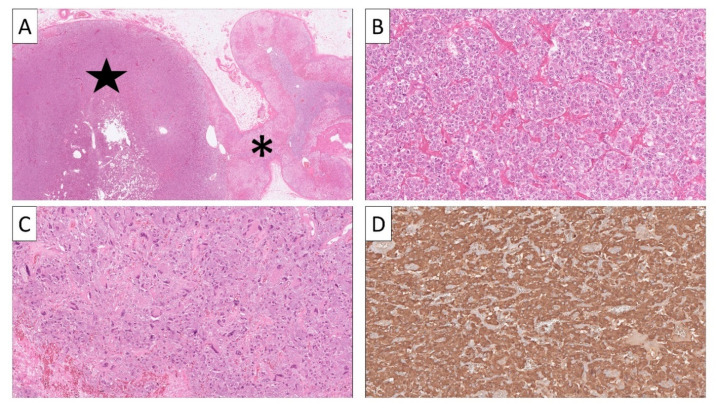
All images represent staining with hematoxylin-eosin (H&E) unless otherwise specified. (**A**) Low-power image of a pheochromocytoma (star) arising in adrenal gland asterisk). (**B**) High-power view illustrating the classical nested appearance of the tumor cells with a finely granular, amphophilic cytoplasm. (**C**) Subsets of cases may display nuclear pleomorphism and hyperchromatic nuclei. (**D**) Immunohistochemistry reveals diffuse chromogranin A positivity.

**Figure 4 cancers-14-00917-f004:**
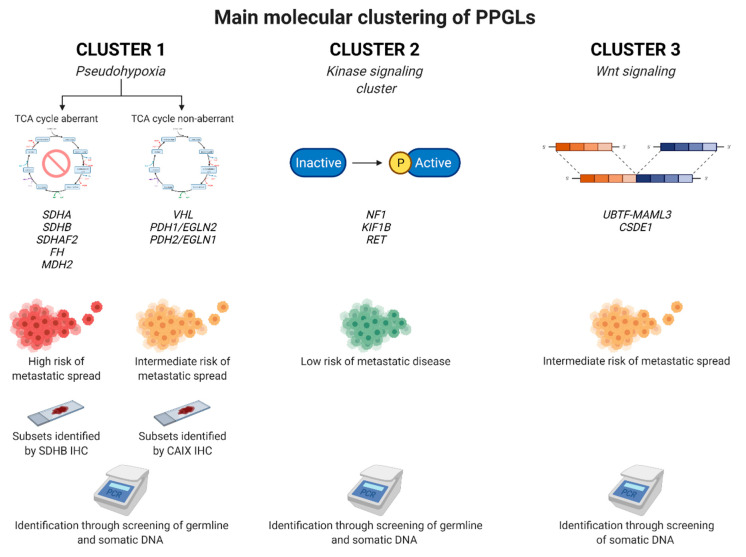
Main molecular clustering of pheochromocytomas and paragangliomas (PPGLs). Tumors are traditionally divided into four main transcriptomal clusters depending of the mRNA profiles, of which three are detailed here with a representative set of specific genes highlighted for each sub-group. In terms of clinical importance, cluster 1 tumors exhibit the highest proportion of metastatic cases and are usually driven by somatic or constitutional mutations in genes responsible for the cellular response to hypoxia. These gene variants either aggregate in genes encoding enzymes propelling the tricarboxylic acid (TCA) cycle, or in genes regulating hypoxia-inducible factor more directly (“TCA cycle non-aberrant”). While this triaging is helpful, it should be stressed that individual genes of certain sub-groups may have a risk of dissemination that does not fit perfectly with the assigned cluster. Subsets of cases adhering to the cluster1 sub-groups may be identified by immunohistochemical (IHC) analyses targeting the SDHB and CAIX proteins. Cluster 2 is defined by PPGLs exhibiting mutations in genes regulating kinase-associated pathways, and these tumors usually have low metastatic potential. Finally, cluster 3 is represented by PPGLs driven by *MAML3* gene fusions or *CSDE1* mutations, causing an aberrant Wingless type (Wnt) pathway signaling. These tumors have an intermediate risk of metastatic disease. Created with BioRender.com.

**Figure 5 cancers-14-00917-f005:**
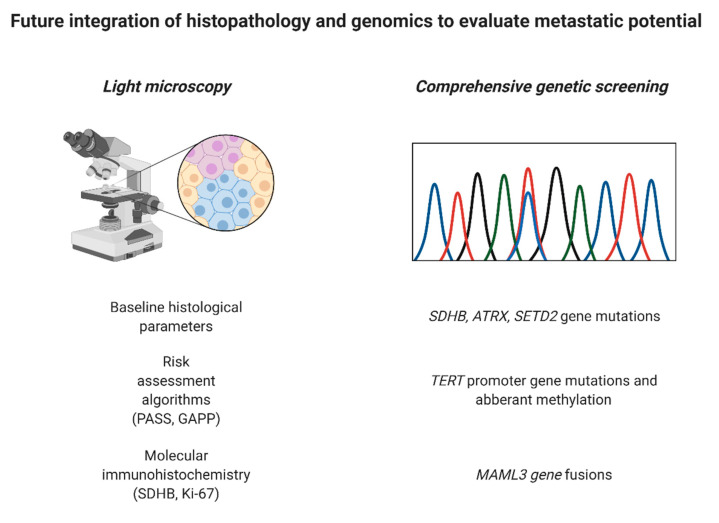
Potential future work-up of pheochromocytomas and abdominal paragangliomas may require a combination of histology and molecular immunohistochemistry as well as screening for somatic genetic aberrations in order to facilitate the detection of cases with the potential to metastasize. Created with BioRender.com.

**Figure 6 cancers-14-00917-f006:**
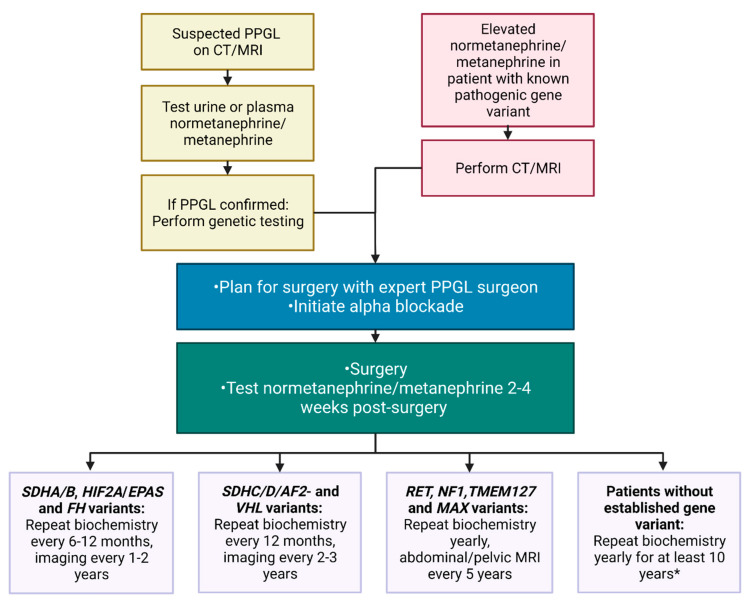
Proposed clinical flowchart for clinical management of PPGL patients. It is recommended to individualize management due to factors such as age and aggressiveness of disease, which are factors not accounted for in this scheme. When interpreting normetanephrine/metanephrine levels, any drug effect must be excluded. *Recurrences may be detected after more than 10 years, so some departments advise lifelong annual follow-up with biochemical tests. Created with BioRender.com.

**Table 1 cancers-14-00917-t001:** Outpatient pre-procedural medical management.

Class of Medication	Medication Name	Approach to Titration	Dosing	Monitoring/Goals of Therapy	Side Effects and Counseling
Alpha-adrenergic blockade	Phenoxybenzamine Doxazosin	Start at least 10–14 days prior to procedure. Titrate daily based on orthostatic blood pressure.	Starting dose: usually 10 mg once or twice daily, gradually increased. Final dose varies (60–120 mg total daily dose in divided doses). Starting dose: usually 1 mg once or twice daily, or 4 mg once daily, gradually increased. Final dose varies (6–40 mg) * total daily dose in divided doses.	Monitoring includes: daily orthostatic vitals, side effects. The goal is low normal blood pressure.	Fatigue, lightheadedness, tachycardia, nasal congestion, diarrhea Counseling: Optimal hydration Increase salt intake. Avoid driving if lightheaded.
Beta-adrenergic blockade	Propranolol Metoprolol succinate Atenolol	Start 3–7 days prior to procedure. Start after alpha-adrenergic blockade. Titrate daily based on heart rate.	Starting dose: 10 mg every 6–8 h, gradually increased. Final dose varies (30-90 mg total daily dose). Starting dose: 25 mg daily. Final dose varies (50–200 mg) daily in divided doses. Starting dose: 25 mg daily. Final dose varies.	Absence of tachycardia, with a baseline heart rate <80–90 beats/minute	Usually none if started after alpha-adrenergic blockade and close monitoring as well as treatment of short duration.
Calcium channel blockade	Amlodipine	Usually used as an additive agent when blood pressure is uncontrolled with alpha- and beta-blockade.	Starting dose: 5 mg, increase to 10 mg if needed.	Monitoring includes blood pressure measurements.	Usually none with close monitoring and treatment of short duration.
Catecholamine synthesis inhibitor	Metyrosine	Usually used when inadequate or intolerant to alpha blockade, when difficult resection is anticipated. Titrated based on the Mayo Clinic protocol. Day 1: 250 mg every 6 h Day 2: 500 mg every 6 h Day 3: 500 mg every 6 h Day 4: 750 mg every 6 h Day 5: 1000 mg every 6 h, last dose of 1000 mg on the morning of procedure		Monitor for side effects	Fatigue Sedation Dizziness Depressed mood Diarrhea, anorexia Extrapyramidal side effects Counseling: Optimal hydration. Avoid driving. Contact physician if extra-pyramidal side effects occur.

* Forty mg is a very high dose of doxazosin that is not usually employed except in rare cases. In most settings, the maximal dosage of doxazosin employed is 16 mg, and calcium-channel blockers or metyrosine are employed if control is still inadequate. On the other hand, in some patients a final dose lower than 6 mg (e.g., 2 mg or 4 mg) is sufficient to reach the blood pressure target.

## Data Availability

Not applicable.

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
