# Peer review of "Pheochromocytomas and Abdominal Paragangliomas: A Practical Guidance"

_cancers, 2022, doi:10.3390/cancers14040917_

Round 1

Reviewer 1 Report

Calissendorff et al. present a well-structured, informative review. I have only minor comments:

  • General comment: the review focuses on pheos and abdominal In the introduction the authors state that they will not discuss HNPGL in this review. It would be helpful to briefly discuss the different locations where PPGL may arise and explain why they focus on abdominal PPGLs and what they mean by abdominal PPGL. It remains currently unclear whether pelvic or thoracic PPGL are included/excluded? Also in the main text, it is sometimes confusing as they use the general abbreviation “PPGL”, but they only refer to pheos and abdominal paragangliomas, e.g. line 139: “..hypertension is present in up to 95% of PPGLs..” or line 178: “..PPGLs are rarely non-functional..”. This would not be correct if all PPGL (+HNPGL) are included. I suggest using “pheochromocytomas and abdominal paragangliomas” instead or the authors could make clear in the beginning that in this specific review the abbreviation PPGL refers only to pheos and abdom. PPGLs.

  • Introduction: for completeness, please define the term pheochromocytoma

  • Biochemical diagnosis: the authors mention that the biochemical profile depends on the hereditary syndrome (line 170). I think it would be nice to make the discussion a bit broader by explaining how molecular characteristics influence the biochemical phenotype (incl. cell differentiation and enzyme activity according to underlying mutation/cluster).

  • Imaging: Line 221: the “high accuracy of F-FDG-PET in SDHB negative tumors” does not correspond to the given sensitivity of 62%.

  • Genetics: Lines 293-300: I suggest adding a reference (Fishbein et al. 2017)

  • Management: Line 370: In metastatic disease surgery is not always the first step. I would therefore change to “..for most patients with PPGL”.

  • Table 1: This table is currently difficult to read. The layout could be changed for better readability (landscape format? rows more clearly separated? less text as repetition with main text?)

  • Metastatic disease: Lines 488-500: I think SSTR-based PRRT is an important and emerging therapeutic option in PPGL and recommended as first-line treatment in slowly/moderately progressive metastatic PPGL, especially cluster 1 tumours (Lenders et al. J. of Hypertension 2020). The value of SSTR-based PRRT could be elaborated in more detail.

  • Typos: line 16 discreet, line 20 afflicted, line 34 was, line 184 methanephrines

Author Response

Reviewer 1:

Calissendorff et al. present a well-structured, informative review. I have only minor comments:

  • General comment: the review focuses on pheos and abdominal In the introduction the authors state that they will not discuss HNPGL in this review. It would be helpful to briefly discuss the different locations where PPGL may arise and explain why they focus on abdominal PPGLs and what they mean by abdominal PPGL. It remains currently unclear whether pelvic or thoracic PPGL are included/excluded? Also in the main text, it is sometimes confusing as they use the general abbreviation “PPGL”, but they only refer to pheos and abdominal paragangliomas, e.g. line 139: “..hypertension is present in up to 95% of PPGLs..”or line 178: “..PPGLs are rarely non-functional..”. This would not be correct if all PPGL (+HNPGL) are included. I suggest using “pheochromocytomas and abdominal paragangliomas” instead or the authors could make clear in the beginning that in this specific review the abbreviation PPGL refers only to pheos and abdom.

Our response: Thank you for this comment. We added the following phrase to line 79: In this review, PPGLs denote pheochromocytomas and abdominal paragangliomas, i.e., lesions inferior to the diaphragm, in which paragangliomas originate from the sympathetic chain ganglia in the pelvis, abdomen or thorax (Figure 1).

                       Figure 1 is novel and describes the anatomical location for PPGLs. The Figure text reads:

                      Figure 1. Anatomic overview of pheochromocytoma and abdominal parganglioma. While pheochromocytoma arise in the adrenal medulla, abdominal parganglioma derive from sympathetic paraganglia. The latter entity most commonly occurs in the retroperitoneum along the sympathetic

                       trunk, not seldom within the organs of Zuckerkandl. Paragangliomas          

                       may also develop in paraganglia distributed along the urinary and gastrointestinal tracts. Not depicted herein are rare paragangliomas of the

                       cauda equina region. Created with BioRender.com.

                       Further, it is true that the references about hypertension is concerning pheochromocytomas, this has been changed (line 145).

                     We have also clarified that “Pheochromocytomas and abdominal paragangliomas are rarely non-functional…”

  • Introduction: for completeness, please define the term pheochromocytoma

Our response: We have defined this in the revised Introduction section: “Pheochromocytomas are adrenal tumors arising from chromaffin cells of the adrenal medulla…”

  • Biochemical diagnosis: the authors mention that the biochemical profile depends on the hereditary syndrome (line 170). I think it would be nice to make the discussion a bit broader by explaining how molecular characteristics influence the biochemical phenotype (incl. cell differentiation and enzyme activity according to underlying mutation/cluster).

Our response: We have now elaborated on this matter (new text in italics): For example, patients with VHL/SDHx mutant tumors (cluster 1) usually display lower plasma metanephrines than kinase-drived PPGLs (NF1/RET). This may be due to the fact that cluster 1 and 2 tumors differ in expression of the Phenylethanolamine N-methyltransferase (PNMT) enzyme responsible for the conversion of norepinephrine to epinephrine. Cluster 1 noradrenergic tumours (VHL/SDHx driven) therefore show little/absent PNMT, with strong increases of normetanephrine with no or relatively small increases in metanephrine, while cluster 2 adrenergic tumours (driven by kinase-associated mutations, for example NF1/RET) show tumor-derived increases in plasma metanephrine. (Eisenhofer G, Clin Biochem Rev. 2017 Apr;38(2):69-100).

  • Imaging: Line 221: the “high accuracy of F-FDG-PET in SDHBnegative tumors” does not correspond to the given sensitivity of 62%.

Our response: We agree and now clarify (in italics): 18F-FDG is also less precise in RET-related pheochromocytomas, but has high accuracy for SDHB positive and but less for SDHB negative tumors, with 83% and 62% sensitivity, respectively

  • Genetics: Lines 293-300: I suggest adding a reference (Fishbein et al. 2017)

Our response: We have added the reference (Fishbein L et al. 2017).

  • Management: Line 370: In metastatic disease surgery is not always the first step. I would therefore change to “..for most patients with PPGL”.

Our response: We have clarified this aspect with the following edit in italics: Surgical resection is the mainstay of therapy for any patient with a localized PPGL.

  • ‘Table 1: This table is currently difficult to read. The layout could be changed for better readability (landscape format? rows more clearly separated? less text as repetition with main text?)

Our response: We agree that the readability could be improved. We have now edited the table accordingly.

  • Metastatic disease: Lines 488-500: I think SSTR-based PRRT is an important and emerging therapeutic option in PPGL and recommended as first-line treatment in slowly/moderately progressive metastatic PPGL, especially cluster 1 tumours (Lenders et al. J. of Hypertension 2020). The value of SSTR-based PRRT could be elaborated in more detail.

Our response: We agree that PRRT is an important tool in metastatic PPGL thus we now add (in italics) “ 177Lu-DOTATATE [123-125] often being used and these are the recommended choices depending on which shows best expression [6]. DOTATATE therapy might be slightly more effective than 131I-MIBG [126], especially in cluster 1 TCA cycle aberrant tumors (Taieb D, Endocr Relat Cancers 2019)”.

  • Typos: line 16 discreet, line 20 afflicted, line 34 was, line 184 methanephrines

Our response: The spelling has been corrected, the division of afflicted has been changed. The word “was” is not in line 34. Line 187, metanephrines is now spelled correct.

Reviewer 2 Report

consider a figure showing common extra-adrenal locations of abdominal PGL, and a bit more discussion of whether these can mimic metastatic disease from adrenal pheo,  I suggest a bit more discussion on differential diagnosis and treatment of metastatic pheo versus primary PGL.

Tables and figures appropriately serve their purpose. I suggest adding a figure of common abdominal locations of primary PGL.

the statement that greater malignancy with SDHB mutation is caused by increased epigenetic alterations is not sufficiently well established for such a strong statement of causality.

Author Response

Reviewer 2:

Consider a figure showing common extra-adrenal locations of abdominal PGL, and a bit more discussion of whether these can mimic metastatic disease from adrenal pheo,  I suggest a bit more discussion on differential diagnosis and treatment of metastatic pheo versus primary PGL.

Our response: In the revised manuscript, we now write in the Imaging section: In targeted adrenal investigations, the differential diagnosis lipid-poor adrenal cortical adenoma can most often be ruled out, as pheochromocytomas exhibit Hounsfield units > 10. Adrenal metastases from extra-adrenal origin or the occurrence of an adrenal cortical cancer can however not be safely distinguished with CT.

We also added the following to the section Metastatic disease: Differential diagnosis are other malignancies as lymphomas and metastasis from breast cancer.

Tables and figures appropriately serve their purpose. I suggest adding a figure of common abdominal locations of primary PGL.

Our response: We have now added an illustration following the Introduction, describing abdominal locations of PPGLs (Figure 1).

The statement that greater malignancy with SDHB mutation is caused by increased epigenetic alterations is not sufficiently well established for such a strong statement of causality

Our response: We have edited this statement: …which in turn may activate hypoxia inducible factor alpha (HIF-alpha) and may interfere with the epigenetic machinery

Reviewer 3 Report

This is a review article aimed to aimed to give "a practical guidance" in the management of patients with pheochromocytoma and abdominal paragangliomas. The manuscript is of interest as it reviews a relatively rare disease whose management has experienced significant changes in the last years thanks to the new discoveries especially regarding genetic profiling. The main strength of the manuscript is the completeness of the review that indeed includes sections on epidemiology, clinical presentation, biochemical diagnosis, imaging, histology, genetics, and it also investigates the management of this disease in rare settings such as pregnancy.

However, I believe there are some limitations that need to be addressed before the article can be considered for publication. The main limitation is that many of the sections give only a general overview and therefore some of the “practical guidance” at the end seems vague. This is especially true for sections “3.7 Management”, “3.8 Follow-up” and “3.9 Metastatic disease”. For example, in the follow-up section it would be useful to add some recommendation regarding patients that are asymptomatic mutation carriers (without personal history of PPGL).

Moreover, by reading the title the authors wanted to address specifically “abdominal paragangliomas” but in the text it is not really clear which indications are specific for abdominal rather than head-and-neck paraganglioma. Another main limitation is that the english is sometimes difficult to read (for example: lines 15-16, lines 221-214….).

Minor issues:

  • Line 34: please replace “highlight” with “highlighting”
  • Line 66: please add “abdominal” to “current knowledge of PPGL” if this is really the focus
  • Lines 80-88 are difficult to read. The authors present results where the incidence increase was explained (at least mainly) by an increase in imaging studies performed, but they speculate that there could also be a true incidence increase. They should provide at least a brief discussion supporting this speculation.
  • Line 136: “severe cardiovascular complications can lead to hypertensive crisis”. Is this sentence correct or was it supposed to be the other way around?
  • Lines 140, 141: Although this is true for most patients with phechromocytoma, there is a relevant proportion of patients with noradrenergic PPGL who experience constant hypertension without paroxysmal symptoms. Please rephrase and be more specific.
  • Lines 158, 159: I would remore "equally" as it is clearly stated that no head-to-head comparison is available and a recent meta-analysis showed higher sensitivity and specificity for plasma tests.
  • Line 174, 175: “a value > 4 times…”. The review that is cited states that a value >2 times the upper limit of normal is suggestive for a functional tumor.
  • Line 182: “CgA are..” please replace with “CgA is…”
  • Please try to improve and reorganize the paragraph including lines 211-232 as it sounds repetitive and difficult to read.
  • Table 1, Doxazosin “final dose varies (6-40 mg)”: this may be misleading and “practical guidance” difficult to follow. Indeed, 40 mg is a very high dosage that is not usually employed except in rare cases and it should be better specified at least in the test. In most settings the maximal dosage of doxazosine employed is 16 mg and calcium-channel blockers or metyrosine are employed if control is still inadequate. On the opposite, in some patients a "final dose" lower than 6 mg (ex. 2 mg or 4 mg) is sufficient to reach the blood pressure target.
  • Lines 407-409: This sentence should probably be moved to another paragraph. It seems out of context.
  • Lines 478-480: This is still a matter of debate and in my opinion the authors should be more cautios in this statement (See: 10.1016/j.jamcollsurg.2013.04.027).
  • Line 490: please add “slowly” before “progressing”.

Author Response

Reviewer 3

This is a review article aimed to aimed to give "a practical guidance" in the management of patients with pheochromocytoma and abdominal paragangliomas. The manuscript is of interest as it reviews a relatively rare disease whose management has experienced significant changes in the last years thanks to the new discoveries especially regarding genetic profiling. The main strength of the manuscript is the completeness of the review that indeed includes sections on epidemiology, clinical presentation, biochemical diagnosis, imaging, histology, genetics, and it also investigates the management of this disease in rare settings such as pregnancy.

However, I believe there are some limitations that need to be addressed before the article can be considered for publication. The main limitation is that many of the sections give only a general overview and therefore some of the “practical guidance” at the end seems vague. This is especially true for sections “3.7 Management”, “3.8 Follow-up” and “3.9 Metastatic disease”. For example, in the follow-up section it would be useful to add some recommendation regarding patients that are asymptomatic mutation carriers (without personal history of PPGL).

Our response: We have now added, e.g., some recommendation regarding patients that are asymptomatic mutation carriers in the Follow-up section but also in the other sections.

Moreover, by reading the title the authors wanted to address specifically “abdominal paragangliomas” but in the text it is not really clear which indications are specific for abdominal rather than head-and-neck paraganglioma. Another main limitation is that the english is sometimes difficult to read (for example: lines 15-16, lines 221-214….).

Our response: We have now tried to clarify that the review specifically addresses abdominal paraganliomas and not head-and-neck ones. We have also tried to improve the English as suggested.

Minor issues:

Line 34: please replace “highlight” with “highlighting”

Our response: This has been altered accordingly.

Line 66: please add “abdominal” to “current knowledge of PPGL” if this is really the focus

Our response: We have added “abdominal” as suggested.

Lines 80-88 are difficult to read. The authors present results where the incidence increase was explained (at least mainly) by an increase in imaging studies performed, but they speculate that there could also be a true incidence increase. They should provide at least a brief discussion supporting this speculation.

Our response: We have now briefly discussed this as suggested.

Line 136: “severe cardiovascular complications can lead to hypertensive crisis”. Is this sentence correct or was it supposed to be the other way around?

Our response: Thank you for noting this. The sentence has now been rephrased to be the other way around.

Lines 140, 141: Although this is true for most patients with phechromocytoma, there is a relevant proportion of patients with noradrenergic PPGL who experience constant hypertension without paroxysmal symptoms. Please rephrase and be more specific.

Our response: We agree and the sentence has now been rephrased accordingly.

Lines 158, 159: I would remore "equally" as it is clearly stated that no head-to-head comparison is available and a recent meta-analysis showed higher sensitivity and specificity for plasma tests.

Our response: “equally” has been removed as suggested.

Line 174, 175: “a value > 4 times…”. The review that is cited states that a value >2 times the upper limit of normal is suggestive for a functional tumor.

Our response: This has been corrected as suggested.

Line 182: “CgA are..” please replace with “CgA is…”

Our response: This has been changed as suggested.

Please try to improve and reorganize the paragraph including lines 211-232 as it sounds repetitive and difficult to read.

Our response: The section has now been rephrased.

Table 1, Doxazosin “final dose varies (6-40 mg)”: this may be misleading and “practical guidance” difficult to follow. Indeed, 40 mg is a very high dosage that is not usually employed except in rare cases and it should be better specified at least in the test. In most settings the maximal dosage of doxazosine employed is 16 mg and calcium-channel blockers or metyrosine are employed if control is still inadequate. On the opposite, in some patients a "final dose" lower than 6 mg (ex. 2 mg or 4 mg) is sufficient to reach the blood pressure target.

Our response: Yes, the dose of alpha-blockade varies between patients and centers. We have now made some adjustments as suggested.

  • Lines 407-409: This sentence should probably be moved to another paragraph. It seems out of context.

Our response: The sentence has now been rephrased.

  • Lines 478-480: This is still a matter of debate and in my opinion the authors should be more cautios in this statement (See: 10.1016/j.jamcollsurg.2013.04.027)

Our response: We have now rephrased the sentence as suggested.

Line 490: please add “slowly” before “progressing”

Our response: “slowly” has been added as suggested.

Round 2

Reviewer 3 Report

I believe the manuscript has been significantly improved by the authors. 

Just a minor comment:

line 221: I have a question for the authors as this concept is not clear to me. Why should it be easy to distinguish between lipid-poor adenomas and pheo using HU? Are the authors referring to lipid-rich adenomas instead?

Author Response

Our response: You are right is should be lipid rich adenomas, we have altered this as susggested.